# Unveiling the Effect of *NCgl0580* Gene Deletion on 5-Aminolevulinic Acid Biosynthesis in *Corynebacterium glutamicum*

**Jian Wu** [†], **Meiru Jiang** [†], **Shutian Kong, Kunqiang Hong, Juntao Zhao, Xi Sun, Zhenzhen Cui, Tao Chen**  **and Zhiwen Wang** *****

Key Laboratory of Systems Bioengineering (Ministry of Education), Frontier Science Center for Synthetic Biology (Ministry of Education), Department of Biochemical Engineering, School of Chemical Engineering and Technology, Tianjin University, Tianjin 300072, China
* Correspondence: zww@tju.edu.cn; Tel.: +86-22-85356605
† These authors contributed equally to this work.

**Abstract:** 5-Aminolevulinic acid (5-ALA) has recently received much attention for its wide applications in medicine and agriculture. In this study, we investigated the effect of *NCgl0580* in *Corynebacterium glutamicum* on 5-ALA biosynthesis as well as its possible mechanism. It was found that the overexpression of *NCgl0580* increased 5-ALA production by approximately 53.3%. Interestingly, the knockout of this gene led to an even more significant 2.49-fold increase in 5-ALA production. According to transcriptome analysis and functional validation of phenotype-related targets, the deletion of *NCgl0580* brought about considerable changes in the transcript levels of genes involved in central carbon metabolism, leading to fluxes redistribution toward the 5-ALA precursor succinyl-CoA as well as ATP-binding cassette (ABC) transporters affecting 5-ALA biosynthesis. In particular, the positive effects of enhanced sugar transport (by overexpressing *NCgl1445* and *iolT1*), glycolysis (by overexpressing *pyk2*), iron uptake (by overexpressing *afuABC*), and phosphate uptake (by overexpressing *pstSCAB* and *ugpQ*) on 5-ALA biosynthesis were demonstrated for the first time. Thus, the transcriptional mechanism underlying the effect of *NCgl0580* deletion on 5-ALA biosynthesis was elucidated, providing new strategies to regulate the metabolic network of *C. glutamicum* to achieve a further increase in 5-ALA production.

**Keywords:** *NCgl0580*; 5-aminolevulinic acid; transporter; transcriptome analysis; *Corynebacterium glutamicum*



## 1. Introduction

5-Aminolevulinic acid (5-ALA) is an endogenous, non-proteinogenic, five-carbon amino acid, which is widely found in microorganisms, plant cells, and animal cells. It is an important precursor for the biosynthesis of heme, chlorophyll, vitamin B12, and other tetrapyrroles [1]. 5-ALA has recently received much attention, primarily due to its great contribution to the treatment of different cancers [2–5] and coronavirus disease 2019 (COVID-19) [6,7], as well as its wide applications as an efficient growth regulator and herbicide in agriculture [8,9]. It is projected that the global 5-ALA hydrochloride market will reach USD 145.7 million by 2028, a significant increase from USD 101.5 million in 2021 [10]. As the market demand for 5-ALA grows, environmentally friendly, inexpensive, and sustainable microbial production of 5-ALA is becoming increasingly popular among researchers compared to chemical synthesis [11].

There are two known natural 5-ALA biosynthesis pathways. One is the C4 pathway found in yeasts, mammals, and some photosynthetic bacteria [1]. In this pathway, 5-ALA synthetase (ALAS) condenses succinyl-CoA and glycine to directly synthesize 5-ALA [12]. The other is the C5 pathway encoded by the key genes *gltX*, *hemA*, and *hemL*, mainly found in some bacteria and plants, which produces 5-ALA from glutamate [1]. However, the C5

pathway is subject to complex feedback regulation by its downstream product heme [13,14] and shows a lower yield compared to the C4 pathway [15]. Several different microbial hosts were used to produce 5-ALA, including *Escherichia coli*, *Corynebacterium glutamicum*, and *Saccharomyces cerevisiae* [15]. Among them, *C. glutamicum* may have a higher potential to produce 5-ALA due to the absence of a glycine cleavage system [16] and better intrinsic acid tolerance.

The development of high-yield strains is crucial for the efficient production of 5-ALA, and a number of metabolic engineering strategies have been developed to construct optimal 5-ALA-producing strains, including the engineering of key enzymes, redistribution of central carbon fluxes toward precursors, regulation of downstream pathways, cofactor engineering, and transporter engineering [15]. For the high-level biosynthesis of 5-ALA, the efficient export of target products from the cell can avoid the accumulation of toxic substances, such as reactive oxygen species generated by 5-ALA, as well as reduce feedback inhibition to some extent [11]. The serine/threonine transporter RhtA was demonstrated to facilitate the efflux of 5-ALA in different recombinant strains [17]. Kang et al. [18] found that overexpression of *rhtA* increased 5-ALA accumulation by 45.9%, firstly identifying this 5-ALA exporter in *E. coli*. Recently, the introduction of a *rhtA* expression cassette using the T7 promoter enhanced the 5-ALA titer from 40 mg/L to 1.6 g/L in *E. coli* [19]. Auto-inducible expression using the two-component system HrrSA in response to extracellular heme was used to dynamically regulate the expression of *rhtA* in recombinant *C. glutamicum* A30, which increased 5-ALA production by 32.8%, reaching 3.16 g/L [20]. Thus, 5-ALA transport also plays an important role in 5-ALA synthesis in *C. glutamicum*. In addition to the introduction of heterologous RhtA as a transport strategy, changing the permeability of the cell wall to facilitate the transport of 5-ALA also yielded excellent results in *C. glutamicum* [21,22]. However, endogenous 5-ALA transport proteins in *C. glutamicum* have not been reported. Recent studies showed that the *NCgl0580* gene, encoding a permease of the drug/metabolite transporter (DMT) superfamily, has significant effects on the efflux of serine, threonine [23], cysteine [24], and β-alanine [25], implying that it is a transporter with broad substrate specificity and great potential for the export of amino acids and derived substances.

In the present work, we first introduced the C4 pathway into *C. glutamicum* and obtained the 5-ALA-producing engineered strain JS. Then, the gene *NCgl0580*, with a positive effect on 5-ALA synthesis in *C. glutamicum*, was screened by BLAST and gene perturbation experiments. Surprisingly, 5-ALA production was significantly enhanced after the knockout of *NCgl0580* (JSD1), resulting in a 2.49-fold increase compared to the control strain JS, implying that *NCgl0580* may have new functions. Subsequently, RNA-seq and experimental validation of key targets were performed on the *NCgl0580* deletion strain JSD1 to explore the potential function of this gene. This study reveals the transcriptional mechanism through which *NCgl0580* knockout affects 5-ALA biosynthesis, contributing to a better understanding of the 5-ALA pathway and providing new strategies for metabolic engineering to increase 5-ALA production.

## 2. Materials and Methods

### 2.1. Strains, Media, and Culture Conditions

All strains used in this study are listed in Table S1. *E. coli* DH5α was used for the construction of plasmids. *C. glutamicum* J0 was used as the starting strain for engineering to overproduce 5-ALA. For plasmid construction, *E. coli* DH5α was cultured in Luria-Bertani broth (10 g/L tryptone, 10 g/L NaCl, and 5 g/L yeast extract) or on LB agar plates (2% *w/v* agar) at 37 °C and 220 rpm. *C. glutamicum* was grown in brain heart infusion (BHI) broth (74.4 g/L) or on BHI agar plates (2% *w/v* agar) for routine culture. For 5-ALA production, *C. glutamicum* was fermented in CGIII medium (21 g/L 3-(4-morpholino) propanesulfonic acid (MOPS), 2.5 g/L NaCl, 10 g/L tryptone, and 10 g/L yeast extract, pH 7.0). For the revival of *C. glutamicum* engineered strains stored at −80 °C, BHI agar plates were streaked and incubated for approximately 48 h. Then, single colonies formed on the plates were

transferred into tubes containing 5 mL of BHI medium and grown at 30 °C and 220 rpm for 24 h, after which 1 mL of the resulting seed culture was used to inoculate 500 mL shake flasks containing 50 mL of CGIII medium with 10 g/L glucose and grown at 30 °C and 220 rpm for 12 h. The resulting secondary seed culture was used to inoculate 500 mL shake flasks containing 50 mL of CGIII medium and 10 g/L glucose to an initial $OD_{600}$ of 0.5 for further fermentation of 5-ALA at 30 °C and 220 rpm. Where indicated, 7.5 g/L of glycine and 0.1 mM IPTG (isopropyl b-D-1-thiogalactopyranoside) were added to the cultures when the $OD_{600}$ reached 5. Chloramphenicol (10 mg/L) and/or kanamycin (25 mg/L) were added where appropriate to promote plasmid retention. All reagents in this study were purchased from Sangon Biotech (Shanghai, China), except for the brain-heart infusion, which was purchased from Hopebio (Qingdao, China).

### 2.2. Construction of Plasmids

All the plasmids and primers used for plasmid construction are shown in Tables S1 and S2. For the construction of 5-ALA-producing strains of *C. glutamicum*, the vector pXPS was constructed. The $P_{sod(C131T)}$ promoter was amplified from the genomic DNA of *C. glutamicum* strain CGS15 [26] by PCR using the primer pair pXPS-1/2. The codon-optimized coding sequence of $hemA_{RP}$ was synthesized by GENEWIZ, Inc. (Table S3). The fragment was amplified using the primer pXPS-3/4. Then, the two fragments were fused by PCR. The resulting product was digested with *Kpn*I and *Eco*RI and ligated between the corresponding sites of pXK. The pXPP, pXPT, and pXPG were constructed analogously. The plasmid pEC-XK99E carrying an IPTG-inducible promoter was used for plasmid-based gene expression. The plasmid pEC-NCgl0580 was constructed to overexpress the *NCgl0580* gene. The *NCgl0580* gene was amplified from the genome of *C. glutamicum* using primers NCgl0580-1/NCgl0580-2, and the resulting PCR product was digested with *Xba*I/*Sda*I and inserted between the corresponding sites of pEC-XK99E. The plasmids pEC-NCgl2065, pEC-NCgl0581, pEC-pyk2, pEC-Ncgl1445, pEC-iolT1, pEC-afuABC, pEC-pstSCAB, pEC-ugpQ, and pEC-aceA were constructed analogously. The two-step homologous recombination method using the suicide vector pD-sacB was applied to genetically modify *C. glutamicum*. To knock out *NCgl0580*, the upstream and downstream fragments of the target gene were amplified from the genome of *C. glutamicum* using the primers pD-NCgl0580-1/2/3/4, respectively, and fused by PCR. The fused fragment was then digested with *Sma*I/*Bam*HI and inserted between the corresponding sites of pD-sacB, obtaining the plasmid pD-NCgl0580. The plasmids pD-NCgl*2065*, pD-NCgl0580TAA, pD-NCgl0581, PD-sdhCAB, and pD-gdhA were obtained by the same method using corresponding primers and restriction enzymes.

### 2.3. Analytical Methods

The concentration of glucose was quantified using an SBA-40E sensor instrument (Institute of Microbiology, Shandong, China). The concentration of cells was assessed by measuring the optical density at 600 nm ($OD_{600}$) using a conventional UV-visible spectrophotometer (TU-1901, PUXI, Beijing, China). The 5-ALA concentration was determined using a modified Ehrlich's reagent [18]. The heme concentration was determined using a fluorescence-based assay [27].

### 2.4. Transcriptome Analysis

Strains JSD1 and JS (control) were cultured in CGIII medium supplemented with 10 g/L glucose and 7.5 g/L glycine at 30 °C and 220 rpm for 12 h to the mid-logarithmic phase. The cells were harvested by centrifuging the fermentation broth for 30 min at 4 °C and 5000 rpm and were washed once with RNase-free water. The collected cells from 6 samples (3 parallel samples for strains JSD1 and JS) were shock-frozen in liquid nitrogen and sent to GENEWIZ (Suzhou, China) for transcriptome sequencing using an Illumina HiSeq platform. The raw image data of the sequencing results were identified by base calling using the software Bcl2fastq (version 2.17.1.14) to obtain the raw sequencing data. The software Cutadapt (version 1.9.1) was used to pre-process the raw data, filter the

low-quality data, and remove contamination and splice sequences to obtain the clean data. The annotated genome (GCF_000011325.1) of *C. glutamicum* ATCC 13032 was used as the reference for comparative analysis of the clean data. Gene expression calculations were performed using the software HTSeq (version 0.6.1), which calculates gene expression using the FPKM (fragments per kilo bases per million reads) method [28]. The analysis of genetic differences was performed using DESeq2 (version 1.6.3) of the Bioconductor package, which is based on a model with a negative binomial distribution. Pathway enrichment analysis was performed using KEGG pathways as a unit, and hypergeometric tests were applied to identify pathways in which the differentially expressed genes are significantly enriched compared to the whole genomic background.

### 2.5. Real-Time Quantitative PCR (RT-qPCR)

RT-qPCR was further applied to ensure the validity of the RNA-seq data. The cDNA was reverse-transcribed from total RNA obtained from GENEWIZ (Suzhou, China) and synthesized using a TransScript first-strand cDNA synthesis SuperMix (TransGen, Beijing, China) with random primers. The real-time fluorescence quantitative RT-qPCR was performed on a LightCycler® 480 instrument (Roche, Basel, Switzerland), which was used to amplify and quantify the PCR products using TransStart® Top Green qPCR SuperMix (TransGen, Beijing, China). The 16S ribosomal RNA of *C. glutamicum* was adopted as an internal reference. The fold change in transcript levels was calculated by the comparative $2^{-\Delta\Delta Ct}$ method. All primers used for RT-qPCR are listed in Table S4. The RT-qPCR of the target genes was conducted in parallel with triple replicates.

## 3. Results and Discussion

### 3.1. Enhancement of 5-ALA Synthesis by Overexpressing NCgl0580

Transport engineering plays an essential role in strain development by enabling more efficient export of target products to avoid cytotoxicity as well as feedback inhibition from excessive product accumulation inside the cell [15]. Since similar functions of homologs of *E. coli* exporters have been shown in *C. glutamicum* as reported before [23], we speculated that homologs of RhtA in *C. glutamicum* may also be involved in the transport of 5-ALA. In this study, the amino acid sequence of RhtA from *E. coli* was used to find homologs in *C. glutamicum* by BLAST, obtaining seven candidate genes. The top two proteins encoded by the genes *NCgl0580* and *NCgl2065* with the highest similarity were selected as potential 5-ALA transporters in *C. glutamicum*.

To identify their effects on 5-ALA biosynthesis, a 5-ALA-producing chassis strain of *C. glutamicum* was constructed. As the key enzyme of the C4 pathway, 5-ALA synthase (ALAS, encoded by *hemA*) directly affects 5-ALA synthesis in terms of the enzyme's intracellular expression level, enzyme activity, and feedback inhibition by heme (Figure S1). Here, the ALAS mutant C75A/R365K (*hemA*$_{RP}$) from *Rhodopseudomonas palustris* [29] was utilized for 5-ALA synthesis because of its naturally higher enzymatic activity and excellent resistance to heme inhibition. Subsequently, the strong promoters $P_{glyA}$, $P_{pgk}$, $P_{tuf}$ [30], and $P_{sod(C131T)}$ [26] were selected to ensure the high-level expression of 5-ALA synthase in *C. glutamicum*. The codon-optimized coding sequence of *hemA*$_{RP}$ was synthesized and expressed in *C. glutamicum* under the control of the above promoters using the plasmid pXK, resulting in the engineered strains JP, JT, JG, and JS, respectively. The results showed that $P_{sod(C131T)}$-driven *hemA*$_{RP}$ expression was the most effective for 5-ALA production in *C. glutamicum*, as the corresponding strain accumulated 0.59 ± 0.01 g/L of 5-ALA, which was significantly higher than in the other strains (Figure 1A), without negative effects on cell growth and sugar consumption (Figure 1B). The engineered strain *C. glutamicum* JS was used as the chassis for the functional study of the genes *NCgl0580* and *NCgl2065* by perturbing their expression.

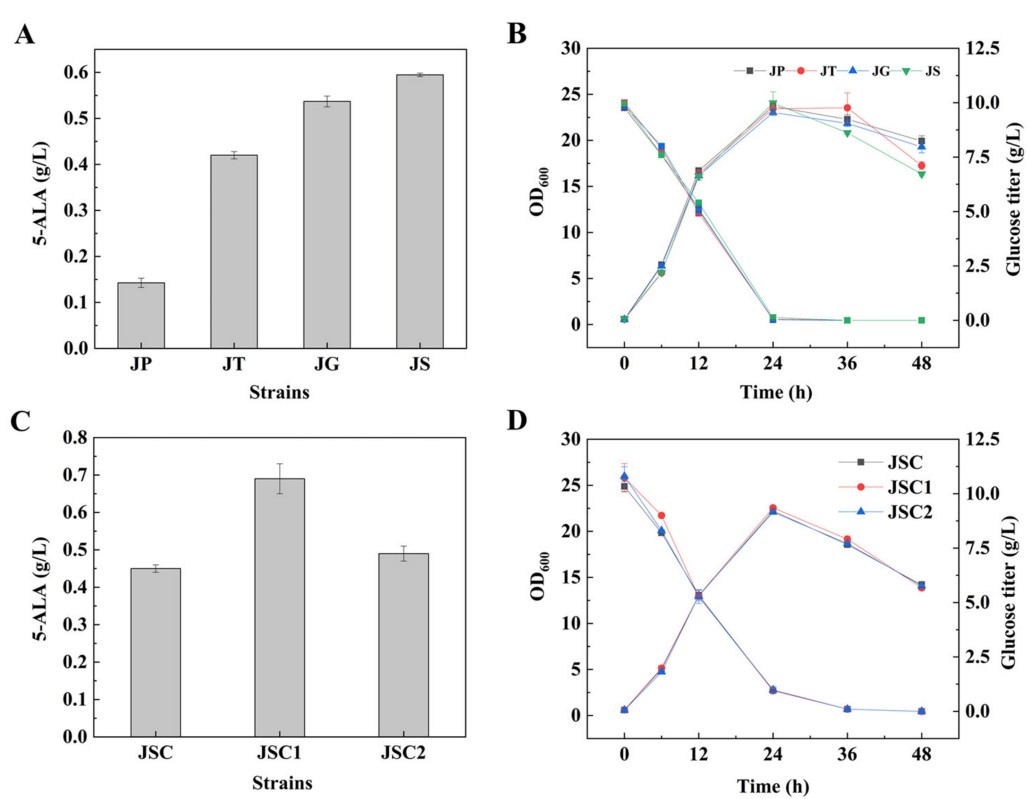

**Figure 1.** 5-ALA titer (**A**), cell growth, and sugar consumption (**B**) of strains JP, JT, JG, and JS overexpressing *hemA$_{RP}$* from pXK-based plasmids under the control of promoters $P_{pgk}$, $P_{tuf}$, $P_{glyA}$, and $P_{sod(C131T)}$, respectively. 5-ALA titer (**C**), cell growth, and sugar consumption (**D**) of *NCgl0580* and *NCgl2065* overexpression strains JSC1 and JSC2, as well as JSC (control). Error bars represent the standard deviations from three independent experiments.

Strains JSC1 and JSC2 were obtained by overexpressing the genes *NCgl0580* and *NCgl2065* in strain JS, respectively, using the IPTG-inducible medium-copy plasmid pEC-XK99E as the vector backbone. After 48 h of shake-flask fermentation, the 5-ALA titers of strains JSC1 and JSC2 reached 0.69 ± 0.04 g/L and 0.49 ± 0.02 g/L, respectively, representing increases of 53.3% and 8.9% compared to the control strain JSC (0.45 ± 0.01 g/L) harboring the empty plasmid pEC-XK99E (Figure 1C). The 5-ALA titer of strain JSC was lower than that of strain JS, probably due to the effects of antibiotic stress and the cytotoxicity of IPTG [31]. As seen in Figure 1D, cell growth and sugar consumption were not affected by the overexpression of *NCgl0580* and *NCgl2065*. Overexpression of *NCgl0580* resulted in a significant increase in the 5-ALA titer compared with overexpression of *NCgl2065*. The protein encoded by the *NCgl0580* gene, annotated as the permease of the DMT family in the *Kyoto Encyclopedia of Genes and Genomes (KEGG)* [32], was screened in this study as a homolog of RhtA (*E. coli*) and showed similar functions to RhtA that export serine and threonine in previous studies [17,23]. Thus, based on the results that overexpression of *NCgl0580* increases the 5-ALA titer, it is speculated that the protein encoded by *NCgl0580* may exhibit the same function as RhtA for the transport of 5-ALA.

### 3.2. Unexpected Effect of NCgl0580 Deletion on 5-ALA Biosynthesis

To further determine the roles of *NCgl0580* and *NCgl2065* in 5-ALA transport, these two genes were deleted in *C. glutamicum*, resulting in strains JSD1 and JSD2, respectively. The results showed that the deletion of *NCgl2065* (JSD2) produced a slight increase in the 5-ALA titer. Strikingly, the *NCgl0580* knockout strain JSD1 exhibited a 2.49-fold increase in the 5-ALA titer to 2.13 ± 0.03 g/L (Figure 2A). This result was not consistent with the expectation that the knockout of the transport protein would cause a decrease in the

production of the target compound. In addition, plasmid-based overexpression of *NCgl0580* in the JSD1 strain decreased 5-ALA production, reverting to the original phenotype to some extent (Figure S2). It therefore stands to reason that *NCgl0580* may play additional roles in *C. glutamicum*.

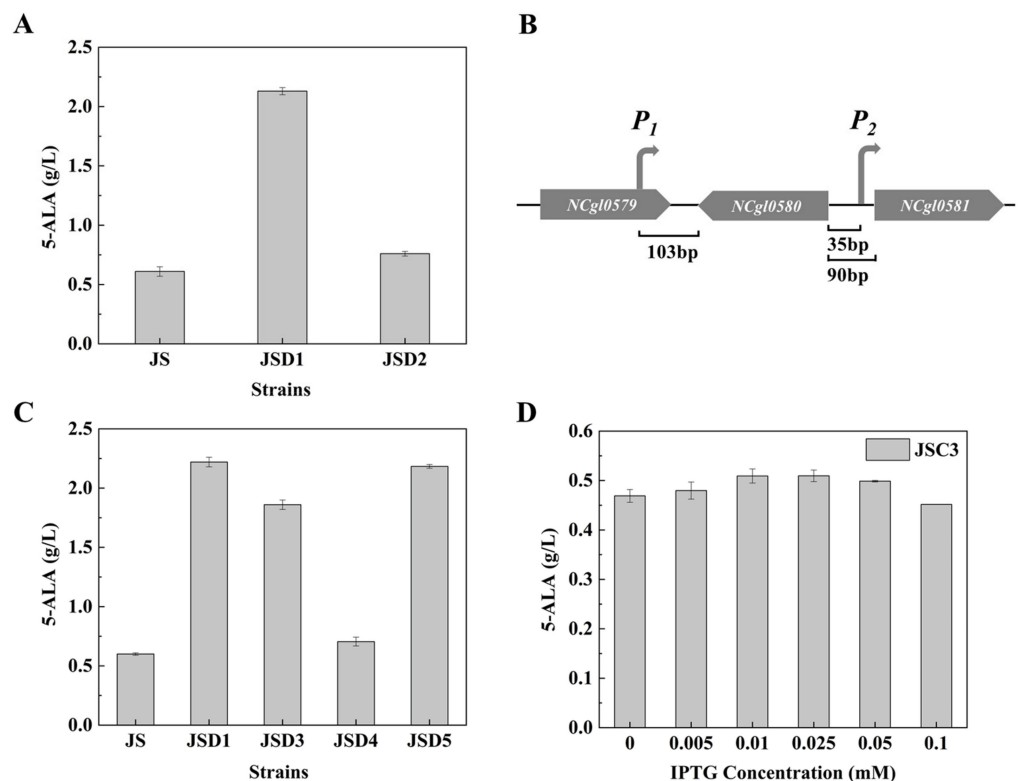

**Figure 2.** Effects of *NCgl0580*, *NCgl2065*, and *NCgl0581* perturbation on 5-ALA biosynthesis. (**A**) 5-ALA titers of the *NCgl0580* and *NCgl2065* deletion strains, JSD1 and JSD2, compared to JS (control). (**B**) Distribution of genes up- and downstream of *NCgl0580* and predicted promoters (50 bp) in the genome of *C. glutamicum*. The predicted promoters *P1* and *P2* received scores of 0.96 and 0.65, respectively, from NNPP, which gives output scores between 0 and 1. A higher score indicates a more accurate prediction. (**C**) 5-ALA titer of strain JSD3 with three TAA termination codons inserted before the 10th, 18th, and 27th amino acids of *NCgl0580* and *NCgl0581* deletion strain JSD4, as well as the *NCgl0580* and *NCgl0581* double knockout strain JSD5, and control strains JS and JSD1. (**D**) 5-ALA titer of *NCgl0581* overexpression strain JSC3 induced with different isopropyl thiogalactoside (IPTG) concentrations. The error bars represent the standard deviations from three independent experiments.

To confirm that the high level of 5-ALA biosynthesis was caused by *NCgl0580* knockout, we investigated the functions of genes upstream and downstream of *NCgl0580* on the genome. Gene *NCgl0581* was observed to be adjacent to *NCgl0580* (Figure 2B) and was reported to encode a global transcriptional regulator in *C. glutamicum* [33]. In addition, a putative promoter *P1* was predicted by Neural Network Promoter Prediction (NNPP) of the Berkeley Drosophila Genome Project (BDGP) [34] (https://www.fruitfly.org/seq_tools/promoter.html (accessed on 30 December 2022)) between the genes *NCgl0579* and *NCgl0580* (Figure 2B). Thus, we speculated that this promoter may act directly on the latter gene, *NCgl0581*, after *NCgl0580* knockout, leading to changes in the expression intensity of the transcriptional regulator *NCgl0581* and consequent changes in 5-ALA biosynthesis. To investigate whether the significant increase in 5-ALA production was caused by the altered expression level of *NCgl0581* or the deletion of *NCgl0580* itself, three stop codons were inserted at the 5′ end of *NCgl0580* in strain JS to terminate its translation, avoiding the effect on *NCgl0581* gene expression due to the knockout of *NCgl0580*. The results showed that 5-ALA production was significantly elevated (Figure 2C), which implied

that the substantial increase in 5-ALA biosynthesis after the knockout of *NCgl0580* may be related to the inactivation of *NCgl0580* itself. To exclude the effect of *NCgl0581* on 5-ALA synthesis, the expression of *NCgl0581* was directly perturbed to investigate its effect on 5-ALA biosynthesis. The knockout of *NCgl0581* did not cause significant changes in 5-ALA biosynthesis in strains JS and JSD1 (Figure 2C). In addition, the overexpression of *NCgl0581* from the plasmid pEC-XK99E with the inducible $P_{trc}$ promoter did not cause significant changes in 5-ALA biosynthesis in strain JS when induced with different IPTG concentrations (Figure 2D). Therefore, it can be concluded that the *NCgl0580* knockout caused a significant increase in 5-ALA production. Thus, the mechanism through which the deletion of *NCgl0580* affected 5-ALA biosynthesis required further investigation.

### 3.3. Transcriptomic Analysis of the Effect of NCgl0580 Knockout on 5-ALA Synthesis

As already noted, the deletion of *NCgl0580* was found to significantly increase 5-ALA biosynthesis. To better understand the underlying mechanism, whole-transcriptome sequencing was performed to assess the transcriptional response of the high-producing strain JSD1. The complete raw RNA-seq data are shown in Table S5. A total of 952 genes showed significant changes ($|\log_2 \text{ratio}| \geq 1$, q-value (fdr and padj) $\leq 0.05$) in their transcription levels between JS and JSD1 (505 up- and 447 downregulated), representing over 30.9% of the *C. glutamicum* genome (Figure 3A). Real-time quantitative polymerase chain reaction (RT-qPCR) was performed to identify changes in the expression of the 28 genes involved in representative processes and functions exhibiting high concordance with the RNA-seq data (Table S6). The results showed that the *NCgl0580* knockout affected the expression of many genes. Zakataeva et al. [35] pointed out that cellular metabolites and their derivatives exported by the proteins of the DMT family (RhtA et al.) may play an important role in global regulation as the signal compounds involved in the intracellular communication triggering quorum sensing (QS) systems [36]. Many of these proteins are characterized by wide substrate specificity and are involved in exporting amino acids, purines, and other metabolites, which may play a signaling role [35]. Based on the previous description, we know that the protein encoded by *NCgl0580* belongs to the DMT family and has high homology and functional similarity to RhtA. We therefore speculated that the knockout of *NCgl0580* affects the normal efflux of signal molecules and interferes with intercellular signaling communication, thereby causing a systematic rearrangement of the cellular metabolism.

KEGG is the main public database for pathway annotation [32]. The metabolic and signal transduction pathways associated with the differentially expressed genes (DEGs) were determined by KEGG pathway enrichment analysis. Figure 3B shows the distribution of genes with significant differences between the experimental group and the control group in pathways (Table S7). They were categorized into five groups: metabolism, human diseases, genetic information processing, environmental information processing, and cellular processes. Metabolism and environmental information processing (ABC transporters) contained the majority of differentially expressed genes and were subjected to further in-depth study.

### 3.4. Redistribution of Central Carbon Fluxes toward Succinyl-CoA-Enhanced 5-ALA Biosynthesis

Based on the RNA-seq results, many DEGs were involved in different pathways of central carbon metabolism, including glucose transport, glycolysis, the glyoxylate cycle, the TCA cycle, as well as the glutamate biosynthesis pathway. As shown in Figure 4, the collective expression of genes related to glucose transport and glycolysis was differentially upregulated compared to the control strain. The *NCgl1445* gene, encoding a putative major facilitator superfamily (MFS) transporter potentially associated with the import of sugar based on PTS [37], was upregulated 4.18-fold. The *iolT1* gene was initially reported to encode an inositol transporter [38], and it was later shown that it might encode a PTS-independent glucose uptake system [39]. Notably, *iolT1* was upregulated 2.30-fold. Furthermore, *glK2* (encoding glucokinase), *pfkA* (encoding phosphofructokinase-1), *pyk*

and *pyk2* (both encoding pyruvate kinase), which are the rate-limiting enzymes involved in glycolysis, were upregulated from 1.10 to 2.42-fold. The resulting enhanced sugar transport and glycolysis will increase the rate of sugar catabolism, which is consistent with the accelerated sugar consumption of JSD1 (Figure S3). To investigate the effect of the above genes on 5-ALA biosynthesis, they were overexpressed using the plasmid pEC-XK99E. Compared with JSC, the 5-ALA titer of the overexpression strain increased by 14.7%, as well as by 8.5% after overexpression of *NCgl1445* and *pyk2* (JSCZ1 and JSCZ3). Notably, the *iolT1* overexpression strain JSCZ2 produced $0.69 \pm 0.02$ g/L 5-ALA, representing a 47.2% increase (Figure 5A). It was reported that the PTS system is the primary mechanism of glucose uptake in *C. glutamicum* but may limit the ability of the organism to overproduce compounds with phosphoenolpyruvate (PEP) as a distal precursor [39]. Recent studies reported that the overexpression of the PTS-independent glucose uptake system (encoded by *iolT1*) in *C. glutamicum* may overcome these limitations [39]. For example, the biosynthesis of L-arginine [40], L-lysine [41], L-serine [42], the blue pigment indigoidine [43] (from glutamate), and β-alanine [25] were positively affected by overexpressing *iolT1*. At the same time, it was shown that the carbon fluxes from glycolysis to oxaloacetate may be channeled mainly through *ppc*-encoded phosphoenolpyruvate carboxylase rather than *pyc*-encoded pyruvate carboxylase [21] in *C. glutamicum* when the acetate synthesis pathway is knocked out. This implied that the enhancement of non-PTS glucose import would contribute to sugar uptake and increase the carbon fluxes toward oxaloacetate in *C. glutamicum*, which could effectively enhance 5-ALA synthesis. Enhancing the carbon fluxes from glycolysis into the TCA cycle is an effective strategy for increasing 5-ALA biosynthesis [21]. Therefore, it was speculated that the simultaneous enhancement of the PTS and non-PTS sugar uptake systems increased the glucose uptake capacity of the strain while saving PEP, and the enhanced glycolysis further increased the carbon fluxes into the TCA cycle, resulting in improved 5-ALA biosynthesis.

In the TCA cycle, the *aceA* gene involved in the glyoxylate pathway was upregulated 1.9-fold, while *sdhCAB* was downregulated from 2.53 to 4.18-fold and *sucCD* was upregulated from 1.19 to 1.71-fold (Figure 4). Notably, *aceA* encodes isocitrate lyase, which catalyzes the conversion of isocitrate to glyoxylate and succinate, while *sdhCAB* encodes the three subunits of succinate dehydrogenase, which catalyzes the conversion of succinic acid to fumaric acid. Changes in the transcript levels of these genes may increase the carbon fluxes at the succinate node [44,45]. The carbon fluxes were then directed toward succinyl-CoA via the succinyl-CoA synthase complex (encoded by *sucCD*). To test this conjecture, *aceA* and *sucCD* were overexpressed, while *sdhCAB* was knocked out. The results showed that 5-ALA production was enhanced by 16.5% and 15.6% in strain JSCZ4 overexpressing *aceA* and strain JSDZ1 with *sdhCAB* deletion, compared with the control strain, respectively (Figure 5A,B). Miscevic et al. [45] found that enhancing the glyoxylate cycle and blocking the oxidative branch of the TCA cycle can increase the carbon fluxes at the succinate node and ultimately increase 5-ALA biosynthesis. However, the overexpression of *sucCD* showed no effect on 5-ALA production. Choi et al. [46] experimentally confirmed that the production of 4-HB, which has the same precursor succinyl-CoA as 5-ALA, was enhanced by overexpression of *sucCD* under microaerobic conditions in *E. coli*. Additionally, in our previous work, overexpression of *sucCD* under microaerobic conditions in *C. glutamicum* only exhibited a weak effect on 5-ALA synthesis [47]. Therefore, we speculated that the reason why 5-ALA production was not affected by overexpression of *sucCD* under aerobic conditions in this study may be due to the different genotypes of chassis and production conditions. The glutamate dehydrogenase (GDH)-encoding gene *gdhA* was significantly downregulated 2.21-fold (Figure 4). GDH, the enzyme which catalyzes the conversion of α-ketoglutarate (α-KG) to glutamate, links the carbon and nitrogen metabolisms [48]. To assess the effect of *gdhA* downregulation on 5-ALA biosynthesis, *gdhA* was knocked out in JS, resulting in strain JSDZ2. The 5-AlA titer of JSDZ2 increased by 12.7% compared with JS (Figure 5B). Ge et al. [49] reduced glutamate production by knocking out *gdhA* and significantly increased 5-ALA production by introducing the C4 pathway. Thus, downregulation

of *gdhA* might reduce carbon consumption for glutamate biosynthesis and enable more carbon to flow from α-KG to the C4 pathway, which improves the synthesis of 5-ALA.

Based on these findings, it can be inferred that the fluxes redistribution of central carbon metabolism induced by *NCgl0580* knockout in *C. glutamicum* facilitated the flow of carbon toward the C4 pathway precursor succinyl-CoA, which contributed to 5-ALA biosynthesis.

### 3.5. Enhancement of Iron and Phosphate Uptake Improves 5-ALA Synthesis

ATP-binding cassette (ABC) transporters are ubiquitous membrane proteins that couple the transport of diverse substrates across cellular membranes to the hydrolysis of ATP [50]. Many DEGs were enriched in related pathways based on our RNA-seq data. Interestingly, the iron and phosphate transporters were significantly upregulated (Figure 6). In nature, iron is essential for 5-ALA synthesis, and it has a major impact on the aerobic metabolism of cells [51]. Phosphorus (P) is an essential component of living organisms and is closely related to energy and central carbon metabolism [52]. Therefore, we hypothesized that the upregulation of genes related to these processes also affected 5-ALA biosynthesis.

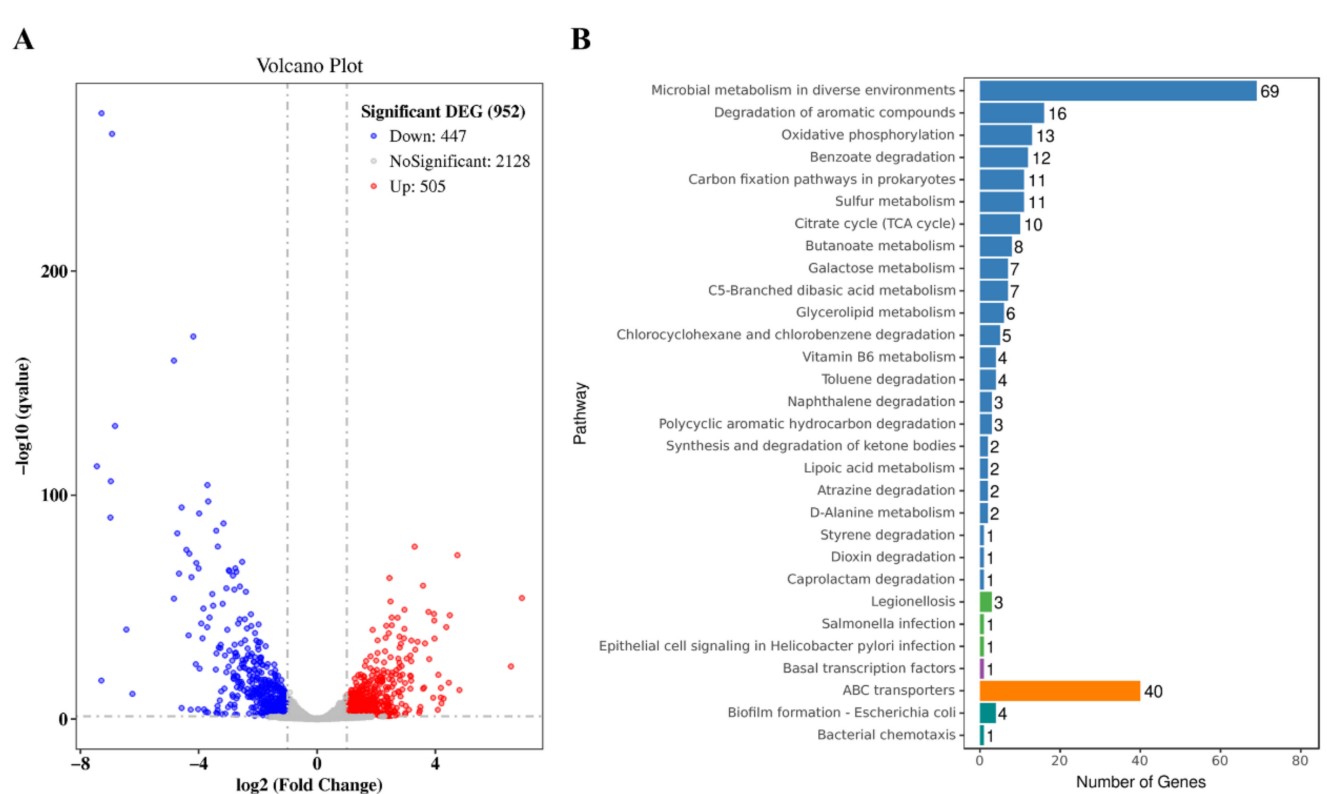

**Figure 3.** Changes in overall gene transcription levels and enrichment of KEGG pathways according to the whole-transcriptome sequencing data. (**A**) A volcano plot of the DEGs. Red and blue dots, respectively, indicate significantly up- and downregulated DEGs; horizontal coordinates represent gene expression fold changes in different samples; and vertical coordinates represent the statistical significance of differences in gene expression. (**B**) Significantly enriched KEGG annotated classification. Blue bars represent metabolism; light green bars represent human diseases; purple bars represent genetic information processing; orange bars represent environmental information processing and dark green bars represent cellular processes. The vertical axis indicates the pathway name, and the horizontal axis indicates the number of genes.

The *afuABC* operon encoding an ABC-type $Fe^{3+}$ transporter was upregulated significantly (from 2.18 to 4.48-fold) (Figure 6). As previously reported, iron plays an important regulatory role in the biosynthesis of 5-ALA. Zhang et al. [53] optimized the iron addition (7.5 mg/L), which resulted in a significant increase in the 5-ALA titer. To investigate the

effect of *afuABC* upregulation on 5-ALA synthesis, *afuABC* was overexpressed, resulting in an increase of the 5-ALA titer by 17.8% (Figure 5A). This result suggested that the overexpression of *afuABC* may promote iron uptake, which in turn enhanced 5-ALA synthesis. To further confirm this conjecture, different concentrations of $Fe^{3+}$ were added to the fermentation medium. The control strain JSC showed the highest 5-ALA production with the exogenous addition of 27 mM $Fe^{3+}$, producing $0.66 \pm 0.01$ g/L of 5-ALA, which was 40.8% higher than without $Fe^{3+}$ addition. The highest 5-ALA production of the *afuABC* overexpression strain was measured in the presence of 18 mM $Fe^{3+}$, with a 5-ALA titer of $0.70 \pm 0.06$ g/L, representing a 26.7% increase compared with no $Fe^{3+}$ addition (Figure 5C). Notably, strain JSCZ5 overexpressing *afuABC* obtained a higher 5-ALA titer at lower concentrations of iron than strain JSC, indicating that the overexpression of *afuABC* enhanced the iron uptake capacity, which eliminated the need for large amounts of extracellular iron addition, improving aerobic cellular metabolism, and consequently promoting 5-ALA biosynthesis.

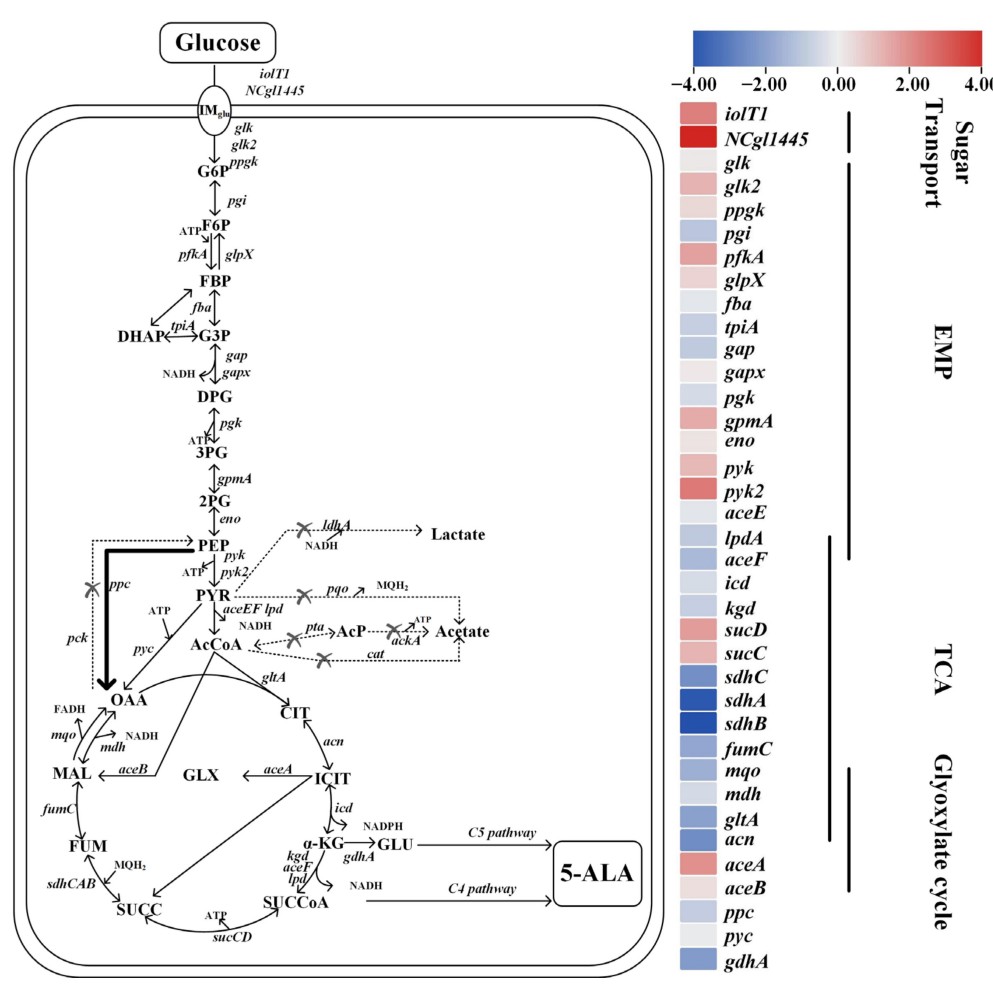

**Figure 4.** Heatmap ($\log_2$-fold change) of the transcription levels of genes involved in central carbon metabolism in JS and JSD1. Shades of red and blue indicate up- and downregulation, respectively.

The transcription levels of *pstSCAB* (encoding an ABC-type Pi uptake system), *ugpQ* (encoding glycerophosphoryldiester phosphodiesterase), and *ugpAEBC* (encoding an ABC-type sn-glycerol 3-phosphate uptake system) were increased more than threefold (Figure 6). The upregulation of these genes seems to promote the uptake of phosphate by the cells [52]. To investigate the effect of the above genes on 5-ALA synthesis, they were overexpressed using the pEC-XK99E vector backbone. The *pstSCAB* and *ugpQ* overexpression strains JSCZ6 and JSCZ7 exhibited 13.4% and 9.9% increases in 5-ALA production, respectively

(Figure 5A). The ABC-type Pi uptake system encoded by *pstSCAB* facilitates the uptake of inorganic phosphate [52]. The glycerophosphodiester phosphohydrolase encoded by *ugpQ* catalyzes the hydrolysis of glycerophosphodiester to sn-glycerol 3-phosphate, which can be utilized as an organic phosphate [54]. In addition, P metabolism is closely related to energy and central carbon metabolism [52]. The interaction between P metabolism and C metabolism is particularly important for amino acid production in *C. glutamicum* because amino acids are derived from intermediates of central carbon metabolism [52]. Here, the overexpression of the transporter may have affected the energy and central carbon metabolism by increasing the uptake of phosphate, thereby enhancing the synthesis of 5-ALA.

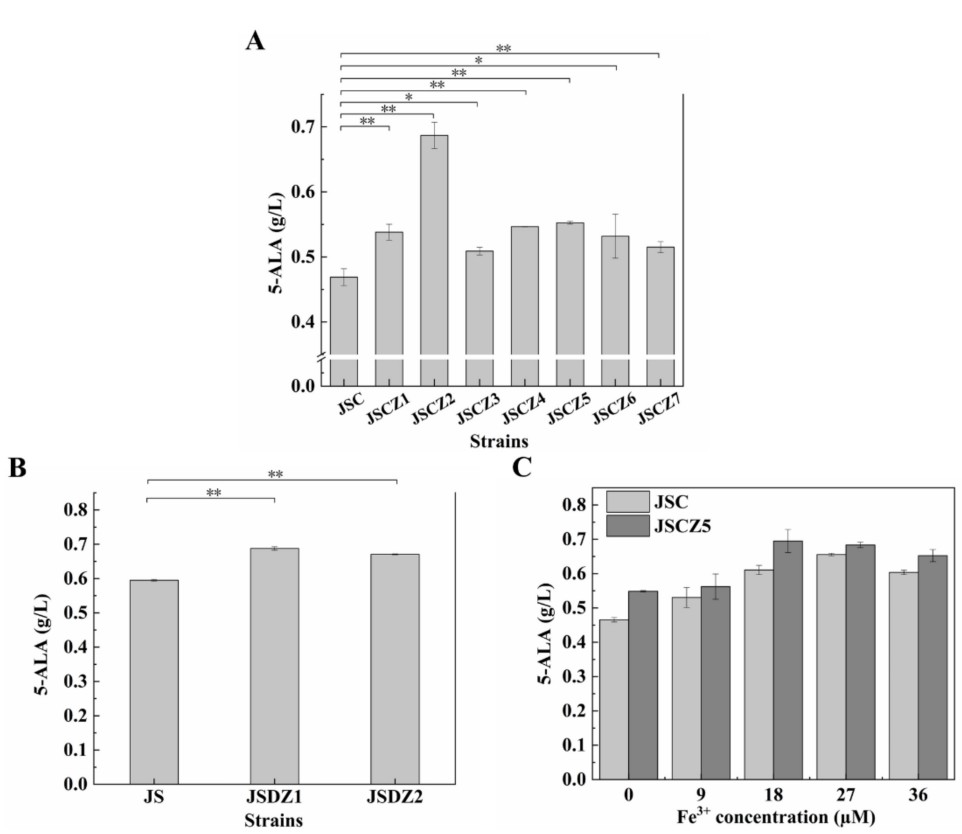

**Figure 5.** Functional validation of key DEGs affecting 5-ALA biosynthesis in JSD1. (**A**) The effects of overexpression of individual genes including *NCgl1445* (JSCZ1), *iolT1* (JSCZ2), *pyk2* (JSCZ3), *aceA* (JSCZ4), *afuABC* (JSCZ5), *pstSCAB* (JSCZ6), and *ugpQ* (JSCZ7) on 5-ALA biosynthesis compared to JSC. (**B**) The effects of the individual deletions of *gdhA* and *sdhCAB* on 5-ALA biosynthesis compared to JS. (**C**) Effect of iron addition on 5-ALA biosynthesis in the strains JSC and JSCZ5. Iron was initially added to the medium at concentrations of 0, 9, 18, 27, and 36 μM. Error bars represent the standard deviations from three independent experiments. The statistical significance of differences in 5-ALA titers was analyzed using the Student's *t*-test in SPSSAU (https://spssau.com/About_spssau.html (accessed on 30 December 2022)). * $p < 0.05$; ** $p < 0.01$.

### 3.6. Effect of Multiple Gene Expression Perturbation on 5-ALA Synthesis

As the metabolic pathway in microbial metabolism is massively complicated and interconnected, the interaction between gene up- and downregulation would affect the global metabolic fluxes. To investigate the effect of interactions between effective genes in previous single-gene validation experiments on 5-ALA synthesis, the genes *NCgl1445*, *iolT1*, *pyk2*, *aceA*, *afuABC*, *pstSCAB*, and *ugpC* were overexpressed based on the *sdhCAB* knockout strain JSDZ1 and the *gdhA* knockout strain JSDZ2, respectively. As can be seen from Figure 7, overexpression of the above genes enhanced 5-ALA production to different extents after deletion of *sdhCAB* or *gdhA*. In particular, overexpression of *iolT1* had the most

significant increase in 5-ALA production in strains JSDZ1 and JSDZ2 by 37.2% and 38.4% (0.80 ± 0.01 g/L and 0.77 ± 0.03 g/L, respectively). Compared with JSC, the 5-ALA titer of strains DZ1C2 and DZ2C2 increased by 70.5% and 64.1%. The results indicate that the combined expression perturbations of effective genes further enhanced 5-ALA production. Therefore, it is speculated that the above-mentioned genes, whose expression was changed by the deletion of *NCgl0580*, interact to promote global metabolic fluxes toward 5-ALA. The mechanism of *NCgl0580* knockout affecting 5-ALA synthesis can be investigated in depth by establishing a gene-interaction network in the future.

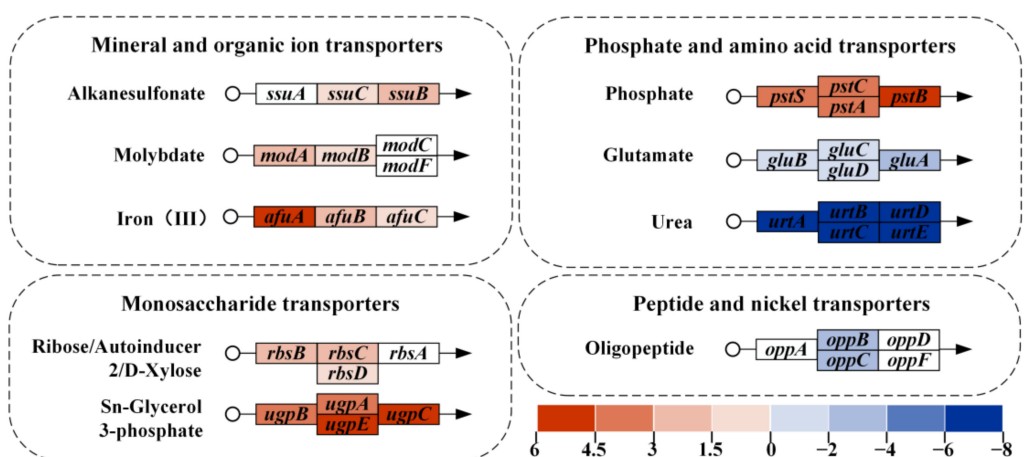

**Figure 6.** Heatmap (log₂-fold change) of transcription levels of genes associated with ABC transporter proteins in the JSD1 strain. Shades of red and blue indicate upregulation and downregulation, respectively.

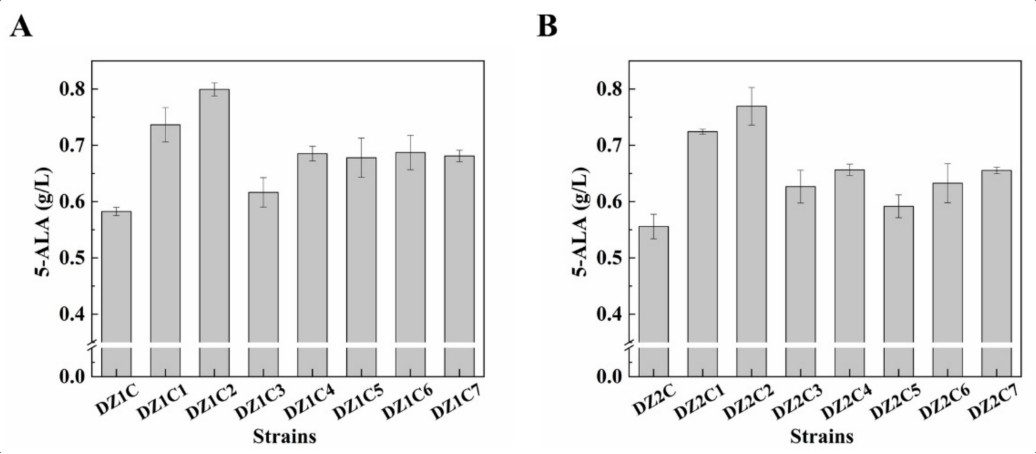

**Figure 7.** Effect of combined expression perturbations of key genes on 5-ALA synthesis. (**A**) The effects of overexpression of genes including *NCgl1445* (DZ1C1), *iolT1* (DZ1C2), *pyk2* (DZ1C3), *aceA* (DZ1C4), *afuABC* (DZ1C5), *pstSCAB* (DZ1C6), and *ugpQ* (DZ1C7) on 5-ALA biosynthesis compared to DZ1C (JSDZ1 harboring pEC-XK99E). (**B**) The effects of overexpression of genes including *NCgl1445* (DZ2C1), *iolT1* (DZ2C2), *pyk2* (DZ2C3), *aceA* (DZ2C4), *afuABC* (DZ2C5), *pstSCAB* (DZ2C6), and *ugpQ* (DZ2C7) on 5-ALA biosynthesis compared to DZ2C (JSDZ1 harboring pEC-XK99E). The error bars represent the standard deviations from three independent experiments.

## 4. Conclusions

In this study, overexpression of *NCgl0580* was demonstrated to facilitate the extracellular accumulation of 5-ALA in *C. glutamicum*. More importantly, the *NCgl0580* knockout resulted in a remarkable 2.49-fold increase in 5-ALA production to 2.13 g/L. Transcriptomic analysis combined with the validation of key DEGs showed that the changes in transcription levels of genes caused by the *NCgl0580* deletion redistributed the fluxes of central

carbon metabolism to enhance the supply of the direct 5-ALA precursor succinyl-CoA. Genes encoding ABC transporters for iron and phosphate uptake were upregulated, improving aerobic metabolism, energy balance, and 5-ALA biosynthesis. Encouragingly, it was confirmed for the first time that overexpression of *NCgl1445*, *iolT1*, *afuABC*, *pstSCAB*, and *ugpQ* could facilitate 5-ALA biosynthesis. This work elucidated the remarkable effect of *NCgl0580* deletion on 5-ALA biosynthesis in *C. glutamicum*, which provides a reference for further study on the transcriptional mechanisms that control 5-ALA biosynthesis. The elucidation of phenotype-related targets also provides a new direction for metabolic engineering to construct high-yield strains.

**Supplementary Materials:** The following supporting information can be downloaded at: https://www.mdpi.com/article/10.3390/fermentation9030213/s1, Figure S1: An overall metabolic pathway for 5-ALA biosynthesis; Figure S2: Effect of plasmid-based *NCgl0580* overexpression in the JSD1 strain on 5-ALA biosynthesis; Figure S3: Determination of cell growth and sugar consumption of strains JS and JSD1; Table S1: Strains and plasmids used in this study; Table S2: Primers used in the construction of plasmids; Table S3: Codon-optimized coding sequence of *hemA* from *Rhodopseudomonas palustris* (*hemA$_{RP}$*); Table S4: Primers used for RT-qPCR; Table S5: The raw data of significant differentially expressed genes between JSD1 and JS; Table S6: The relative transcriptional level of genes in metabolic pathways of *C. glutamicum* JS and JSD1; Table S7: The raw data of pathway enrichment analysis of the differentially expressed genes between JSD1 and JS.

**Author Contributions:** Conceptualization, J.W., M.J. and Z.W.; methodology, J.W., M.J., S.K., X.S. and Z.C.; formal analysis, J.W. and M.J.; investigation, J.W. and M.J.; writing—original draft, J.W.; writing—review and editing, J.W., Z.W., K.H. and J.Z.; project administration, Z.W.; funding acquisition, Z.W. and T.C. All authors have read and agreed to the published version of the manuscript.

**Funding:** This work was financially supported by the National Key Research and Development Program of China (2021YFC2100700) and the National Natural Science Foundation of China (NSFC-22278312).

**Institutional Review Board Statement:** Not applicable.

**Informed Consent Statement:** Not applicable.

**Data Availability Statement:** The data presented in this study are available in the article and Supplementary Materials.

**Conflicts of Interest:** The authors declare no conflict of interest.

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
