# Peer review of "Unveiling the Effect of NCgl0580 Gene Deletion on 5-Aminolevulinic Acid Biosynthesis in Corynebacterium glutamicum"

_fermentation, doi:10.3390/fermentation9030213_

Round 1

Reviewer 1 Report

In this manuscript, metabolic engineering studies on C. glutamicum for 5-ALA were demonstrated. The experimental design and results well support that the strategies provided by the current work are efficient for 5-ALA production. Only some minor aspects deserve to be addressed in order to further improve your submission.

1. Lines 180-181: Please give the exact number of similarities!

2. Lines 182-183: Please provide an overall metabolic pathway for 5-ALA biosynthesis.

3. Lines 189-192: In bacteria including C. glutamicum, it is often observed that the native gene exhibited better performance for product production than the codon-optimized one; please supplement the additional results from the cultivations of C. glutamicum expressing native gene (hemA C75A/R365K from R. palustris) without codon optimization.

4. Line 354: it is stated that the overexpression of sucCD had no effect even though sucCD involve in the conversion from SUCCoA to SUCC which can be competitive with the C4 pathway in 5ALA production.  Additional references or data from previous studies on this are required.

5. Chapter 3.4.: The deletion or overexpression of genes was applied for each of the genes in the experiments. It would be better to contain supplementary results of gene deletion/overexpression of multiple genes to examine the overall gene function in combination.

6. As the metabolic pathway in microbial metabolism is massively complicated and interconnected, the interaction between genes up/downstream would affect the global metabolic flux. However, this study is mostly focused on each of the genes rather than their relationship (e.g., the effect of overexpression of NCgl1445, iolT1, afuABC, pstSCAB, and up) It would be better to include additional materials such as a gene-interaction network for the in-depth analysis.

Author Response

We really appreciate you and the reviewers for the further suggestions on our manuscript
entitled “Unveiling the Effect of NCgl0580 Gene Deletion on 5-Aminolevulinic Acid Biosynthesis
in Corynebacterium glutamicum” (fermentation-2167527). The comments were highly valuable and
very helpful for improving our paper. We thus made a point-to-point response to the questions
raised by the reviewers, as attached below. The revisions were marked in red in the revised main
manuscript.
Should you have any further questions, please let me know. Thank you very much for your
consideration! 

Reviewer 2 Report

The manuscript by Wu et al., describes that the effect of NCgl0580 deletion on 5-aminolevulinic acid production in Corynebacterium glutamicum. The authors first found that the overexpression of NCgl0580 increased 5-ALA production in C. glutamicum. Then they examined the effect of deletion of NCgl0580 on 5-ALA production. Unexpectedly, NCgl0580 knockout led to an even more significant increase of ALA production. To unravel the underlying mechanism of this phenomenon, they carried out the RNA-seq analysis with wild type and NCgl0580 deletion mutant strains. As a result, it was found that the changes in expression levels of genes redistributed the carbon flux to enhance the supply of succinyl-CoA, a precursor of 5-ALA. Subsequently they showed that overexpression of NCgl1445, iolT1, pyk2 and aceA and deletion of sdhCAB increased 5-ALA production. In addition, they also showed that genes involved in iron and phosphate uptake including afuABC, pstSCAB, and ugpQ also affected the 5-ALA production.

The results reported in this manuscript are new and interesting. However, the mechanism how NCgl0580 affects the expression of lots of genes is still unclear.

Major comments

1)     Lines 12-13, Lines 215-16 and Lines 421-422.

The authors clamed that the NCgl0580 encodes a possible ALA exporter. However, there is no experimental evidence showing that the NCgl0580 itself exports 5-ALA. It was only shown that 5-ALA production increased by overexpressing the NCgl0580. These sentences should be changed to avoid the misunderstanding of readers. Or export assay for 5-ALA should be carried out.

2)     Lines 217-261 and Figure 2.

It was shown that NCgl0580 deletion increased 5-ALA production. The authors carefully examined the polar effect of NCgl0580 deletion on expression of the adjacent gene NCgl0581. It is better to carry out the complementation experiment of NCgl0580 deletion mutant by plasmid carrying NCgl0580 to further confirm the effect of NCgl0580 deletion on 5-ALA production.

3)     The mechanism how NCgl0580 affects the expression of lots of genes is still unclear. The authors should discuss about this point.

4)     If the NCgl0580 is not a direct exporter of 5-ALA, it is also mysterious why overexpression and deletion of NCgl0580 both caused increased production of 5-ALA. RNA-seq analysis with a NCgl0580 overproducer should be carried out. The authors should discuss about this point at least.

Minor points

5)     Unfortunately, several figures such as Fig. 3B, Fig. 5 and Fig. 6 are too small to see.

6)     Line 229 and line 234. Ncgl NCgl (capital letter C).

Author Response

We really appreciate you and the reviewers for the further suggestions on our manuscript
entitled “Unveiling the Effect of NCgl0580 Gene Deletion on 5-Aminolevulinic Acid Biosynthesis
in Corynebacterium glutamicum” (fermentation-2167527). The comments were highly valuable and
very helpful for improving our paper. We thus made a point-to-point response to the questions
raised by the reviewers, as attached below. The revisions were marked in red in the revised main
manuscript.
Should you have any further questions, please let me know. Thank you very much for your
consideration

Round 2

Reviewer 2 Report

Comments have been addressed properly.